# An Evaluation of Phylogenetic Workflows in Viral Molecular Epidemiology

**DOI:** 10.3390/v14040774

**Published:** 2022-04-08

**Authors:** Colin Young, Sarah Meng, Niema Moshiri

**Affiliations:** Department of Computer Science & Engineering, University of California San Diego, La Jolla, CA 92093, USA; c2young@ucsd.edu (C.Y.); sjmeng@ucsd.edu (S.M.)

**Keywords:** multiple sequence alignment, phylogenetics, epidemiology, bioinformatics

## Abstract

The use of viral sequence data to inform public health intervention has become increasingly common in the realm of epidemiology. Such methods typically utilize multiple sequence alignments and phylogenies estimated from the sequence data. Like all estimation techniques, they are error prone, yet the impacts of such imperfections on downstream epidemiological inferences are poorly understood. To address this, we executed multiple commonly used viral phylogenetic analysis workflows on simulated viral sequence data, modeling Human Immunodeficiency Virus (HIV), Hepatitis C Virus (HCV), and Ebolavirus, and we computed multiple methods of accuracy, motivated by transmission-clustering techniques. For multiple sequence alignment, MAFFT consistently outperformed MUSCLE and Clustal Omega, in both accuracy and runtime. For phylogenetic inference, FastTree 2, IQ-TREE, RAxML-NG, and PhyML had similar topological accuracies, but branch lengths and pairwise distances were consistently most accurate in phylogenies inferred by RAxML-NG. However, FastTree 2 was the fastest, by orders of magnitude, and when the other tools were used to optimize branch lengths along a fixed FastTree 2 topology, the resulting phylogenies had accuracies that were indistinguishable from their original counterparts, but with a fraction of the runtime.

## 1. Introduction

In molecular epidemiology, phylogenetic analyses are performed on viral sequence data to help reconstruct the evolutionary history and patterns of transmission for highly contagious diseases, providing valuable insights, which may inform intervention strategies [1]. A standard molecular epidemiology workflow typically consists of the following: (1) multiple sequence alignment (MSA), (2) phylogenetic inference, (3) rooting, (4) dating, and (5) transmission clustering. Since each step in the workflow occurs downstream of other analyses, the accuracy of each step is largely dependent on the accuracy of the steps preceding it. Notably, MSA and phylogenetic inference play an integral role in the transmission-clustering techniques utilized in molecular epidemiology. For example, HIV-TRACE is typically considered the best-practice method for Human Immunodeficiency Virus (HIV) transmission clustering: pairwise distances are estimated from an MSA of viral genomes collected from individuals, and two individuals are considered to be epidemiologically “linked” if the pairwise distance between their viral samples is below a user-specified threshold [2]. TreeCluster [3], Cluster Picker [4], and PhyloPart [5] are transmission-clustering tools that separate individuals based on the phylogenetic relationships between viral samples, typically relying on tree topology, branch lengths, and pairwise distances.

Both MSA and phylogenetic inference are NP-Hard computational problems [6], meaning optimal solutions are computationally infeasible on even relatively small datasets. As a result, several heuristic tools have been developed to provide near-optimal approximations. Notably, viral MSA is commonly conducted using MAFFT [7], MUSCLE [8], and Clustal Omega [9], and maximum likelihood viral phylogenetic inference is commonly conducted using IQ-TREE [10], FastTree 2 [11], RAxML-NG [12], and PhyML [13]. As a result, researchers have multiple potential phylogenetic inference workflows at their disposal, each with a theoretical trade-off between runtime and accuracy, yet the impacts of inherent errors and imperfections in these heuristic estimation approaches on downstream molecular epidemiological results are poorly understood.

In this manuscript, we execute multiple commonly used workflows for conducting viral phylogenetic analyses on simulated viral sequence data, modeling HIV, HCV, and Ebola, and we compute multiple methods of accuracy motivated by transmission-clustering techniques.

## 2. Methods

Our simulated datasets were motivated by curated alignments of real-world sequence data; from the Los Alamos National Laboratory (LANL), we obtained full genome MSAs of Ebolavirus (710 total), Hepatitis C Virus (HCV, 471 total), and Human Immunodeficiency Virus Type 1 (HIV-1, 4004 total).

From each curated MSA, we used IQ-TREE to infer a phylogeny under the General Time Reversible (GTR) model of sequence evolution [14] with invariable sites and gamma-distributed site-rate heterogeneity with 20 categories. The GTR model was chosen for this analysis because it is the most general, neutral, independent, finite-sites, time-reversible nucleotide substitution model, and all other neutral, independent, finite-sites, time-reversible nucleotide substitution models are instantiations of the GTR model. From the IQ-TREE results, we obtained the phylogeny, the GTR substitution model parameters, the proportion of invariable sites, and the shape of the gamma distribution. The phylogeny was subsequently rooted using minimum variance rooting [15], and a tree with 100 leaves was subsampled from it. As can be seen in Figure 1, all three phylogenies had a large proportion of internal branch lengths that were near zero, and the Ebola phylogeny also had a large proportion of terminal branch lengths that were also near zero. These parameters were then used to simulate sequence alignments from the subsample of the inferred phylogeny using INDELible v1.03 [16]. Ultimately, 10 replicate datasets were generated for each virus.

We then utilized the true simulated alignment produced by INDELible to evaluate the accuracy of the alignments estimated by MAFFT v7.455, Clustal Omega v1.2.2, and MUSCLE v3.8.31.

To assess the accuracy of phylogenetic inference tools, we ran FastTree v2.1.11, IQ-TREE v1.6.12, RAxML-NG v0.9.0, and PhyML v3.3.20190909 on the true MSAs and compared the results against the tree along which the sequences were simulated. All tools utilized the GTR model of sequence evolution. FastTree 2 and PhyML utilized discrete gamma-distributed site-rate heterogeneity, whereas RAxML-NG and IQ-TREE utilized the FreeRate model [17]. We also utilized IQ-TREE’s ModelFinder Plus functionality [18]. These choices were made in order to run each of the tools in the most generalized inference modes they support (i.e., making the fewest restrictive assumptions *a priori*).

In addition to the above analyses, in which we allowed each phylogenetic inference tool to perform tree search, we also examined a workflow in which a phylogeny is first inferred using FastTree 2, and then each of the other inference tools simply optimizes the branch lengths using the FastTree 2 phylogeny as a fixed tree topology (i.e., the second tool does not execute tree search). Given that FastTree 2 is an order of magnitude faster than the other tools [19], this could potentially be an effective way to reduce computational time for phylogenetic inference workflows.

Lastly, to assess the accuracy of different combinations of tools, we ran each phylogenetic inference method on the sequences aligned by each MSA tool and similarly measured the accuracy of each inferred phylogeny. The metrics used to assess accuracy are described below.

### 2.1. Mean Squared Error

We use Mean Squared Error as a measure of patristic distance matrix similarity
(1)$$\frac{\sum_{i,j}{\left(d_{E}\left(i,j\right)-d_{T}\left(i,j\right)\right)}^{2}}{\left(\begin{array}{c} n\\ 2 \end{array}\right)}$$
where *n* is the total number of sequences and *d_E_*(*i*,*j*) and *d_T_*(*i*,*j*) are respectively the estimated and true pairwise distances between sequences *i* and *j*. Because pairwise distances directly influence the accuracy of transmission-clustering tools, such as HIV-TRACE, Mean Squared Error serves as a valuable indicator for the viability of any given MSA tool. We computed Mean Squared Error on pairwise distances computed directly from estimated MSAs under the Tamura–Nei 93 (TN93) model of sequence evolution [20] using the tn93 component of HIV-TRACE [2], as well as from the pairwise distances along the inferred phylogenies.

### 2.2. Mantel Correlation

We measured the correlation between the estimated and true pairwise distance matrices via the Mantel test [21]. We included both Pearson and Spearman correlation, with 1 indicating perfect correlation (best), −1 indicating perfect anticorrelation, and 0 indicating no correlation.

### 2.3. Robinson–Foulds (RF) Distance

We used both normalized unweighted RF (URF) and weighted RF (WRF) distances to measure the topological difference between two unrooted trees [22], with lower RF distances denoting higher accuracy.

### 2.4. Sum of Pairs (SP) Score

SP Score measures the number of aligned pairs in the inferred MSA that are shared with the true MSA, normalized by the true MSA’s total number of homologies. Higher SP Score indicates higher accuracy. MSA SP Scores were computed using FastSP [23].

### 2.5. Total Columns (TC) Score

TC Score measures the proportion of successfully aligned columns out of the total number of aligned columns in the inferred MSA [23], which ranges from 1 (best) to 0 (worst). The TC Score of each MSA was computed using FastSP.

### 2.6. Compression Factor

Compression Factor is defined as the number of columns in the inferred MSA divided by the number of columns in the true MSA. A perfectly inferred MSA would have a compression factor of 1. MSA Compression Factors were computed using FastSP [23].

## 3. Results

### 3.1. Multiple Sequence Alignment

While Clustal Omega performs poorly on the HCV datasets and MUSCLE performs poorly on the Ebola datasets, MAFFT consistently performs the best across all three viruses, in both runtime and accuracy (Figure 2). Running MAFFT, MUSCLE, and Clustal Omega on the entire curated MSA of 2322 complete HIV-1 genome sequences from LANL took 947 s, 20,343 s, and >86,400 s, respectively. All estimated MSAs of HIV-1 were slightly less accurate than those of HCV and Ebola, with respect to all metrics, aside from Mean Squared Error and Spearman Mantel correlation. Regardless, our results suggest that MAFFT is the best choice of the MSA tools that were tested, in terms of both runtime and accuracy. 

### 3.2. Phylogenetic Inference

Despite the unweighted RF distance of Ebola phylogenies being markedly worse than the others, Ebola phylogenies appear to be comparable or better in accuracy, according to all other metrics (Figure 3). Additionally, using IQ-TREE under the GTR model performs similarly or better than using IQ-TREE with Model Finder Plus (MFP), and with a much lower runtime (Table 1).

However, it must be noted that the simulated MSAs were, themselves, evolved under the GTR model, which gives IQ-TREE GTR an inherent advantage over MFP, which consistently found the best-fit model to be GTR+F+I+G4, i.e., the GTR model with base frequencies empirically computed from the MSA (+F), with invariant sites allowed, with four categories of site-rate heterogeneity under the Gamma model. Phylogenies inferred from the HCV datasets have high Mantel correlations, despite large Mean Squared Error (Figure 3).

As can be seen in Figure 3, FastTree 2 has notably worse accuracy than the other tools, according to nearly all metrics across all three viruses, with the exception of unweighted Robinson–Foulds, in which FastTree performs similarly. However, for each non-FastTree inference tool, the results of simply optimizing the branch lengths of a given FastTree topology have indistinguishable accuracy, with respect to their counterparts, in which tree search is performed by the same tool. Additionally, RAxML-NG performs comparably or better than other tools, in terms of Weighted Robinson–Foulds and Mean Squared Error.

### 3.3. Combinations of MSA and Phylogenetic Inference

As can be seen in Figure 4, as expected, regardless of MSA approach, FastTree 2 consistently yields the lowest-accuracy phylogenies, across all three viruses. The other phylogenetic inference tools seem to perform similarly on HIV-1. Regarding Ebola, however, PhyML, in particular, struggles with Mean Squared Error and Weighted Robinson–Foulds. As for HCV, IQ-TREE and RAxML-NG perform notably better than other tools. Interestingly, even though IQ-TREE with Model Finder Plus consistently converged on GTR+F+I+G4 as the best-fit model, it performed markedly worse than IQ-TREE with GTR+I+R specified; given that +R uses the FreeRate model [17], with four rate categories by default (the same number as +G4), this may suggest that the FreeRate model’s relaxation of the assumption of Gamma-distributed rates yields significantly better trees in this context. Generally, the selection of MSA tool did not have a significant impact on the accuracy of the resulting phylogeny.

### 3.4. Combinations of MSA and Optimized FastTree Topologies

As can be seen in Figure 5, IQ-TREE MFP struggles when inferring a phylogeny from the HCV datasets, having notably worse weighted RF distances and Mean Squared Error than IQ-TREE GTR. In addition, the patristic distance matrices are highly correlated with the true matrix, even in instances where the other metrics may indicate poor performance.

Using the phylogenetic inference tools to optimize the branch lengths on a fixed topology significantly reduces the runtime across the board (Table 1), with the largest reduction in runtime being from RaxML-NG. IQ-TREE (GTR) and IQ-TREE MFP take notably longer just to optimize branch lengths, especially relative to their total runtime. With respect to tree search, FastTree 2 is several orders of magnitude faster than all the other inference tools.

## 4. Discussion

Our results indicate that FastTree 2 consistently runs several orders of magnitude faster than other phylogenetic inference tools. With regard to accuracy, however, FastTree 2 performs worse than the other inference tools, according to nearly all metrics across all three viruses, with the exception of unweighted Robinson–Foulds. This suggests that FastTree 2 is able to infer a reasonably accurate tree topology, but that it poorly estimates branch lengths. These results are consistent with existing benchmarks [19,24]. Interestingly, optimizing the branch lengths along a fixed topology inferred by FastTree 2 results in a phylogeny that is essentially identical to phylogenies inferred solely by a single tool, indicating that this workflow could considerably reduce the runtime of viral phylogenetic inference workflows, with minimal loss in accuracy. It also suggests that each phylogenetic inference tool arrives at roughly equally accurate topologies.

We also observed that phylogenies inferred for Ebola sequences have markedly worse unweighted RF distances than the phylogenies inferred for other viruses. However, the weighted RF distance for Ebola phylogenies actually measures lower. This is because a majority of the branch lengths in the true Ebola phylogeny were near zero (Figure 1), causing the weighting of each unique bipartition to be low, even if there was a significant number of unique bipartitions. The abundance of extremely short branches also intuitively made the phylogeny particularly difficult to infer [25], explaining the high unweighted RF distance. 

Using IQ-TREE under the GTR model of substitution with the FreeRate model of site-rate heterogeneity performed similarly to using IQ-TREE with Model Finder Plus, when used on Ebola and HIV sequences, indicating that Model Finder Plus successfully classifies the evolutionary model in these instances. However, in HCV, IQ-TREE MFP correctly identified the model, transition rates, and proportion of invariable sites, but the relative rates in the model of rate heterogeneity were considerably inaccurate, despite having the correct number of categories, resulting in a noticeable decline in accuracy. This is somewhat surprising; the ModelFinder technique utilized MFP and includes the FreeRate model [17] in its repertoire of models to fit [18]. Upon closer inspection of IQ-TREE’s logs, when testing the GTR model with Gamma-distributed site-rate heterogeneity (+G), MFP tests for both allowed and disallowed invariant sites (i.e., with and without +I), but when testing the GTR model with the FreeRate model of site-rate heterogeneity (+R), MFP does not test for allowed invariant sites (i.e., does not check for +I). Thus, it seems likely that the FreeRate model would yield improved results, but that MFP skews its model testing away from the FreeRate model by disallowing invariant sites and, thus, skewing the inferred rate categories. Nevertheless, MFP also requires significantly more runtime, so unless the user has reason to believe *a priori* that the GTR model with the FreeRate model of site-rate heterogeneity and allowed invariant sites (i.e., GTR+I+R) is a poor fit for their specific virus of interest, the user is best off simply avoiding MFP.

With respect to the original motivation of viral transmission clustering, MAFFT was able to produce MSAs with highly accurate pairwise distances, for all three viruses (Figure 2), and while all phylogenetic inference tools were able to infer phylogenies that had high patristic distance accuracy, all phylogenetic inference tools had high topological error (measured by unweighted Robinson–Foulds distance with respect to “true” tree) when inferring phylogenies from the simulated Ebola MSAs (Figure 3). Because of this, transmission-clustering approaches that rely on pairwise distances (e.g., HIV-TRACE or the non-clade modes of TreeCluster) may be less sensitive to MSA or phylogenetic inference errors than transmission-clustering approaches that rely on the topology of inferred phylogenies (e.g., Cluster Picker, PhyloPart, or the clade modes of TreeCluster). The performance of single-linkage clustering methods based on pairwise distances (e.g., HIV-TRACE or the single-linkage mode of TreeCluster) may be sensitive to subsampling [26], so the non-clade modes of TreeCluster, other than single-linkage mode, may prove to be effective options for transmission clustering, in the face of real-world sampling and inference errors. This idea is consistent with the results of Balaban et al. [3], which benchmarked HIV-TRACE and multiple modes of TreeCluster on a dataset of HIV-1 subtype B *pol* sequences, though more extensive benchmarks of transmission-clustering methods (e.g., across other regions of the HIV genome, across other viruses, etc.) are needed to generalize these observations broadly. 

Based on these results, our recommended workflow to obtain the highest-accuracy MSA and phylogeny with minimal runtime is to (1) use MAFFT to perform MSA, (2) use FastTree 2 under the GTR model with discrete gamma-distributed site-rate heterogeneity to quickly obtain a reasonable tree topology, and (3) use RAxML-NG to optimize branch lengths along the fixed FastTree 2 topology.

## Figures and Tables

**Figure 1 viruses-14-00774-f001:**
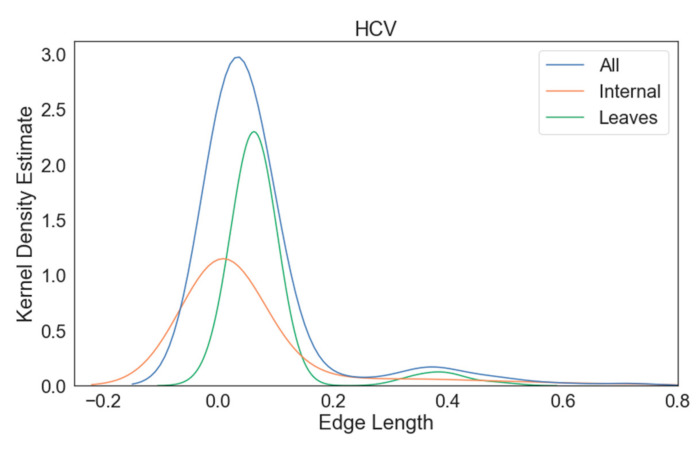
Kernel density estimates of the branch length distributions for the Ebola, HIV, and HCV true phylogenies.

**Figure 2 viruses-14-00774-f002:**
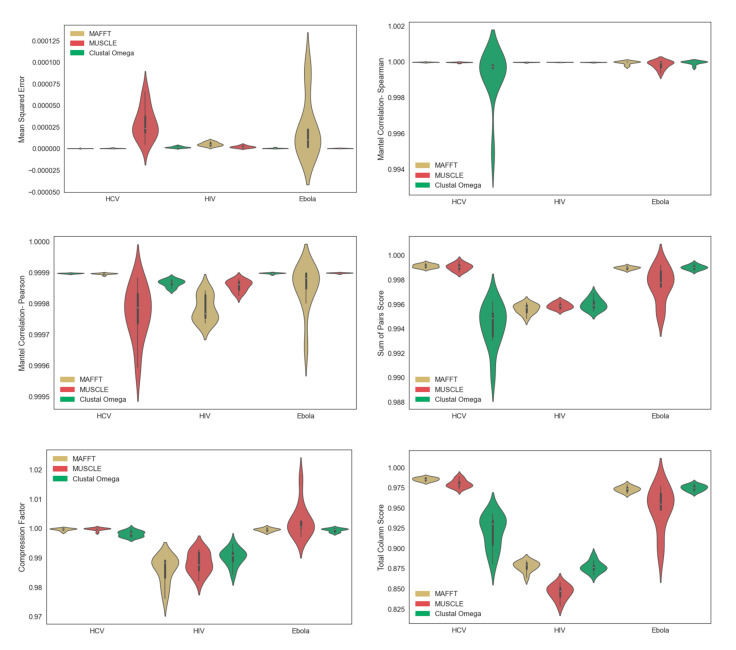
Metrics of sequence alignment accuracy for MAFFT, MUSCLE, and Clustal Omega on 10 simulated replicate datasets of HIV, HCV, and Ebola. Violin plots are shown for Mean Squared Error, Spearman/Pearson Mantel Correlation, SP score, TC score, and Compression Factor.

**Figure 3 viruses-14-00774-f003:**
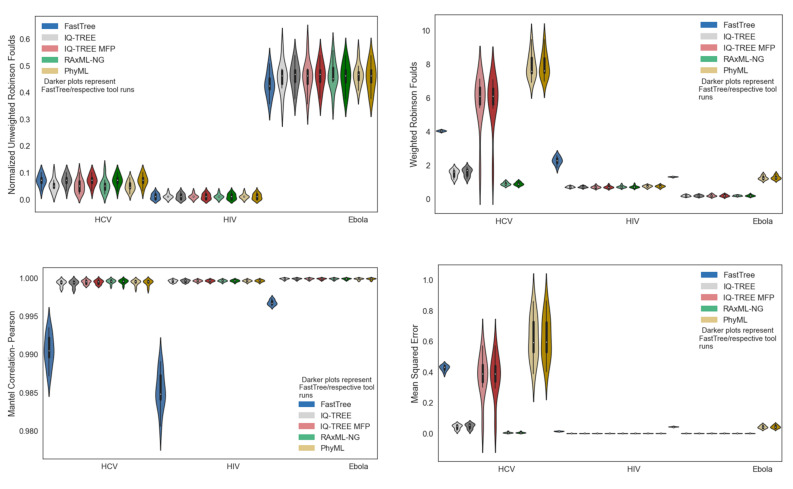
Metrics of phylogenetic inference accuracy for FastTree, IQ-TREE (GTR), IQ-TREE (MFP), RAxML-NG, and PhyML on 10 simulated replicate datasets of HIV, HCV, and Ebola. Phylogenies which result from optimizing branch lengths along FastTree topology are also included. Violin plots are shown for URF, WRF, Pearson Mantel Correlation, and Mean Squared Error. Violin plots showing Spearman Mantel Correlation can be found in Appendix A.

**Figure 4 viruses-14-00774-f004:**
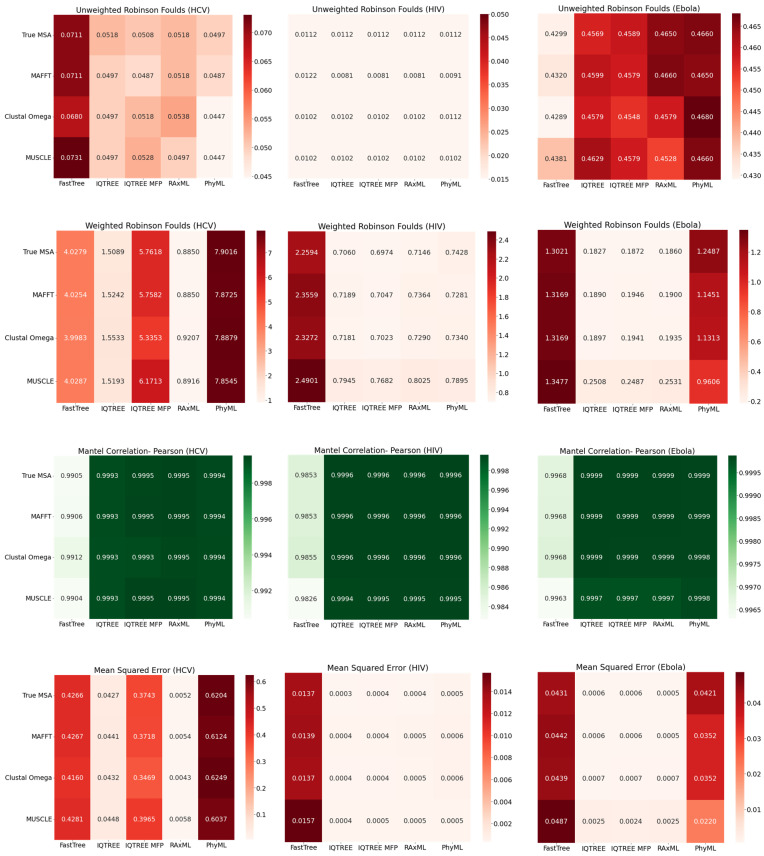
Heat maps comparing the accuracy of phylogenies inferred with FastTree, IQ-TREE (GTR), IQ-TREE (MFP), RAxML-NG, and PhyML from the MAFFT, Clustal Omega, MUSCLE, and true MSAs. Each value of Unweighted Robinson–Foulds (URF), Weighted Robinson–Foulds (WRF), Pearson Mantel Correlation, and Mean Squared Error shown is the average of 10 simulation replicates. Heatmaps showing Spearman Mantel Correlation can be found in Appendix A.

**Figure 5 viruses-14-00774-f005:**
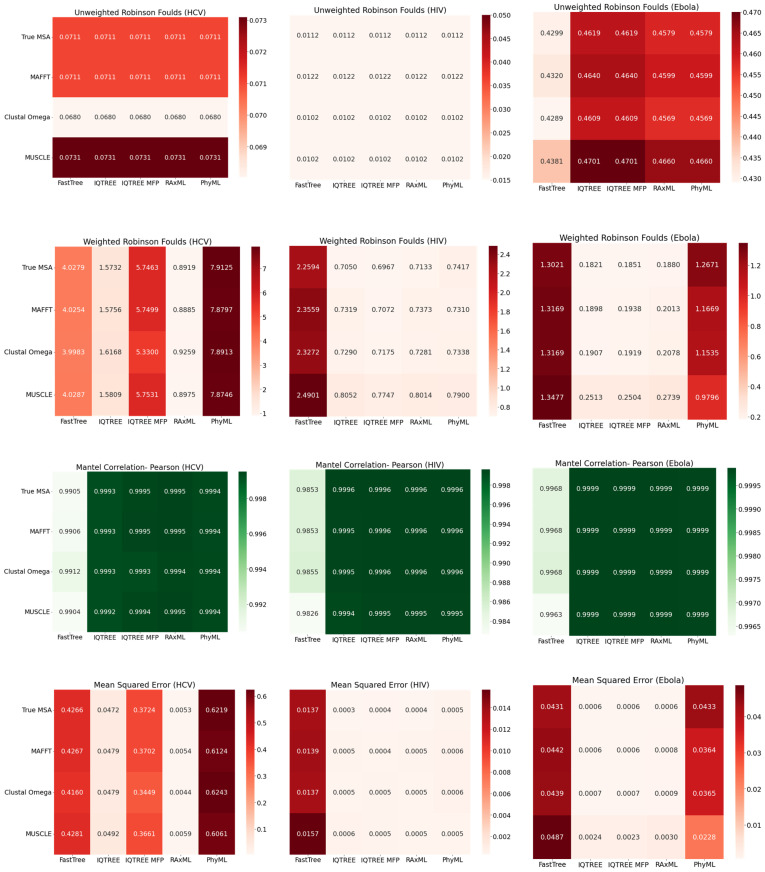
Heat maps comparing the accuracy of FastTree topologies inferred from the MAFFT, Clustal Omega, MUSCLE, and true multiple sequence alignments with branch lengths optimized by IQ-TREE (GTR), IQ-TREE (MFP), RAxML-NG, and PhyML. Each value of Unweighted Robinson–Foulds (URF), Weighted Robinson–Foulds (WRF), Pearson Mantel Correlation, and Mean Squared Error shown is the average of 10 simulation replicates. Heatmaps showing Spearman Mantel Correlation can be found in Appendix A.

**Table 1 viruses-14-00774-t001:** Total runtime for phylogenetic inference (top row) and runtime of branch length optimization on a fixed topology (bottom row) for FastTree 2, RAxML, IQ-TREE (GTR), and IQ-TREE MFP on a curated MSA of 2322 HIV-1 whole genome sequences from LANL. PhyML was unable to execute due to high memory consumption. All runs were executed sequentially on a 4-core 3.5 GHz Intel i5-6600k with 16 GB of memory, and each tool automatically selected an optimal number of threads to use internally.

(Seconds)	FastTree 2	RaxML	PhyML	IQ-TREE	IQ-TREE MFP
Total	645	>604,800	memory	84,931	266,399 (142,286 MFP)
BL optimization	N/A	757	memory	1532	4885

## Data Availability

All raw and processed data, as well as scripts/tools utilized in this study, can be found in the following GitHub repository: https://github.com/Cyoung02/Phylogenetic-Inference-Benchmarking, accessed on 12 November 2020.

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
