# Peer review of "An Evaluation of Phylogenetic Workflows in Viral Molecular Epidemiology"

_viruses, 2022, doi:10.3390/v14040774_

Round 1
Reviewer 1 Report
Particularly, I have some comments for authors:
Introduction
-Lines 46-48. The method of phylogenetic reconstruction associated with the corresponding tool should be exposed.
Methods
-The selection of GTR model of sequence evolution must be explained. Did they base on any literature? Did they previously run any program like Modeltest or Jmodel? In line 79, they also utilized IQ-TREE’s ModelFinder Plus functionality, did they find a different model of sequence evolution? Why didn´t they use it as starting point? Lines 99-100, they also mention that Tamura-Nei 93 model was used in the computation of Mean Squared Error on pairwise distances.
-Lines 85-87 is referring to Table 1, located in Results. This sentence must not be there but in Results or even in Discussion. In addition, Table 1 is then mentioned in line 147, so this table should be located after this text.
-Figure 1 is not explained properly anywhere. It is firstly mentioned in Methods (line 68) without additional explanation. The second mention in the text of Fig.1 (line 134) is not related to such figure; on the contrary, it might refer to figure 2.
-The Mantel test must have a proper reference (Mantel, 1967).
-Line 122: There is a missing dot after 6. (2.6Compression Factor).
Results
-Lines 162 and 163 are referred to Figure 4, but this figure is hosted in the next section. Is this sentence properly used in this subsection (3.2)? Figure 4 is not commented anywhere in the text either (not even in its section 3.3).
-Section 3.3, the acronym PI must previously be explained when firstly mentioned “Phylogenetic Inference”.
-In table 1 is said that PhyML was unable to execute due to high memory consumption with 2,322 HIV sequences. However, this tool was employed in the different tests, wasn´t it? How many HIV sequences were finally used in those tests? 100?
Discussion
This section is short and poorly discussed, without any other references to similar or dissimilar results in literature. In the Introduction, they mentioned several tools (on line or user-friendly), i.e. HIV-TRACE, ClusterPicker, etc. After their analyses, what is the opinion of the authors about these tools? What would be the most accurate?
Supplementary figures
They are not supported without any mention in the text. What do they add?
Reviewer 2 Report
Manuscript is well written and can be consider for publication in this esteemed journal.
I have only one suggestion that authors can add the SARS-CoV-2 data for this MS for Phylogenetic Workflows in a more significant manner.
Reviewer 3 Report
The manuscript “An Evaluation of Phylogenetic Workflows in Viral Molecular Epidemiology” analyse various techniques and methods involved in sequence alignment and phylogenetic analysis. The authors compare the different methods used for the analysis of the viral sequences.
The paper seems to provide detailed information on the different methods used to analyse sequences of different viruses. The authors explained the comparison and gave a summarized outcome to analyse the sequences. However, the author can explain the possible percent of error if he uses only one method to analyse the sequence.
Reviewer 4 Report
This is a very difficult-to-follow manuscript and has required a very slow and attentive reading from this reviewer, almost solely because of the advanced nature of the genomics subject and of the many graphs presented in the manuscript. Without a doubt, a very advanced and critical subject, which I am certain will improve because of the contribution of this manuscript. I have no further comments on the manuscript as it stands.
Round 2
Reviewer 1 Report
The authors have widely improved the manuscript, considering all the suggestions made by the reviewers. However, one of my main concerns related to Discussion is still without solving. Text has added in this section but it remains without discussing properly. In a manuscript, a Discussion section has to include references from the literature that support/ disagree from the results of their study.
